# Bevacizumab Treatment for Patients with *NF2*-Related Schwannomatosis: A Single Center Experience

**DOI:** 10.3390/cancers16081479

**Published:** 2024-04-12

**Authors:** Jules P. J. Douwes, Erik F. Hensen, Jeroen C. Jansen, Hans Gelderblom, Josefine E. Schopman

**Affiliations:** 1Department of Otorhinolaryngology–Head and Neck Surgery, Leiden University Medical Center, 2333 ZA Leiden, The Netherlands; e.f.hensen@lumc.nl (E.F.H.); j.c.jansen@lumc.nl (J.C.J.); 2Department of Medical Oncology, Leiden University Medical Center, 2333 ZA Leiden, The Netherlands; a.j.gelderblom@lumc.nl (H.G.); j.e.schopman@lumc.nl (J.E.S.)

**Keywords:** acoustic neuroma, adverse events, bevacizumab, cerebellopontine angle tumor, hearing, neurofibromatosis type 2, schwannomatosis, tumor response, vestibular schwannoma

## Abstract

**Simple Summary:**

In this study, we explored the use of bevacizumab as a treatment for *NF2*-related schwannomatosis, a condition characterized by the development of schwannomas on both vestibulocochlear nerves. This study revealed that the majority of patients experienced either improved or preserved hearing and effective control of the targeted vestibular schwannoma with bevacizumab. However, common adverse events included hypertension and fatigue, and severe adverse events led to treatment discontinuation in a quarter of the patients. Thus, while bevacizumab demonstrates positive effects on hearing, tumor control, and symptomatology in *NF2*-related schwannomatosis, careful consideration of potential side effects is crucial.

**Abstract:**

(1) Background: *NF2*-related schwannomatosis, characterized by the development of bilateral vestibular schwannomas, often necessitates varied treatment approaches. Bevacizumab, though widely utilized, demonstrates variable effectiveness on hearing and tumor growth. At the same time, (serious) adverse events have been frequently reported. (2) Methods: A single center retrospective study was conducted, on *NF2*-related schwannomatosis patients treated with bevacizumab from 2013 to 2023, with the aim to assess treatment-related and clinical outcomes. Outcomes of interest comprised hearing, radiologic response, symptoms, and adverse events. (3) Results: Seventeen patients received 7.5 mg/kg bevacizumab for 7.1 months. Following treatment, 40% of the patients experienced hearing improvement, 53%, stable hearing, and 7%, hearing loss. Vestibular schwannoma regression occurred in 31%, and 69% remained stable. Further symptomatic improvement was reported by 41%, stable symptoms by 47%, and worsened symptoms by 12%. Treatment discontinuation due to adverse events was observed in 29% of cases. Hypertension (82%) and fatigue (29%) were most frequently reported, with no occurrences of grade 4/5 toxicities. (4) Conclusion: Supporting previous studies, bevacizumab demonstrated positive effects on hearing, tumor control, and symptoms in *NF2*-related schwannomatosis, albeit with common adverse events. Therefore, careful consideration of an appropriate management strategy is warranted.

## 1. Introduction

*NF2*-related schwannomatosis (NF2) is a rare autosomal dominant tumor predisposition syndrome caused by mutations in the NF2 tumor suppressor gene located on chromosome 22. The disease is characterized by the development of bilateral vestibular schwannomas, along with other schwannomas, meningiomas, and ependymomas, in the central and peripheral nervous system [1,2]. The clinical presentation of NF2 is highly variable. While the associated vestibular schwannomas are benign and usually slow growing, they may lead to symptoms such as hearing loss, tinnitus, and vestibular complaints. Without appropriate intervention, the progression of vestibular schwannomas can result in severe cranial nerve deficits, intracranial hypertension, and brainstem compression [2,3].

Most current treatment paradigms for NF2-related vestibular schwannomas consider (micro)surgery, radiotherapy, and pharmacotherapeutic treatment with bevacizumab [4,5,6]. Microsurgery generally achieves satisfactory tumor mass reduction; however, tumor recurrence and hearing loss occur in half of the cases [7]. Radiotherapy offers good tumor control in most NF2 patients but frequently results in hearing loss and other cranial nerve deficits [7,8]. Moreover, the inherent tumor susceptibility of NF2 patients has been linked to a 6% increased risk of malignant transformation or new tumor growth on the irradiated tissues [9]. 

Bevacizumab, a monoclonal antibody, acts to inhibit vascular endothelial growth factor (VEGF) expressed in tumors like vestibular schwannomas [10,11]. Bevacizumab has demonstrated the ability to halt or reverse nervous system tumor progression, particularly in schwannomas. In addition, previous studies have reported improvements in objective and subjective hearing, tinnitus, balance problems, and overall quality of life with bevacizumab treatment [10,11,12,13,14,15,16,17,18]. Bevacizumab has therefore become a widespread treatment for NF2-related schwannomas. Even so, the efficacy and durability of its therapeutic effects can vary between schwannomas and between patients. At the same time, side effects of bevacizumab, including hypertension, proteinuria, and several blood disorders, are frequently reported [18,19]. There is still no international consensus about the optimal treatment dosage, frequency, and duration. Even now that patents have expired, bevacizumab remains an expensive treatment and creates a burden on healthcare costs.

Variable treatment regimens and clinical outcomes, frequent side effects, and cost of bevacizumab treatment were reasons to conduct a retrospective analysis on the clinical effectiveness and adverse effects of bevacizumab treatment for vestibular schwannoma in NF2 patients in our institution.

## 2. Materials and Methods

### 2.1. Participants and Study Design

The present study included NF2 patients eligible for medical treatment with bevacizumab at Leiden University Medical Center (LUMC), a tertiary NF2 referral center in Leiden, The Netherlands, between 1 January 2013 and 31 October 2023. Inclusion criteria comprised a patient age ≥18 years, a confirmed diagnosis of NF2 [20], administration of at least one dose of bevacizumab, and written informed consent. Exclusion criteria comprised insufficient demographic or clinical data.

Following informed consent, we conducted a retrospective, single center, non-blinded data-analysis using electronic patient charts to explore demographics, treatment regimens, and clinical characteristics. A highly secure online database was set up for data storage and analysis. The study protocol was approved by the institutional review board and conducted in full compliance with the ethical standard of the 2013 Declaration of Helsinki.

### 2.2. Treatment Protocol and Outcomes

As per standard-of-care in The Netherlands, 7.5 mg/kg bevacizumab was intravenously administered every three weeks for at least six months or until there was a clinically relevant event, as judged by the treating oncologist and/or NF2 multidisciplinary team. At any time point, treatment with bevacizumab could be discontinued or modified at the discretion of the treating specialist or in accordance with patient preference. Likewise, bevacizumab treatment could be restarted if signs or symptoms required reintervention after a prolonged period of discontinuation.

Hematologic and urinary evaluations were performed prior to the start of bevacizumab treatment and repeated regularly until treatment was discontinued. Vital functions were taken at every admission for bevacizumab treatment. Toxicity was recorded at every visit at the outpatient clinic while treatment was ongoing. In case of an adverse event, this was recorded in the patient chart and reported to the treating specialist. Action was taken if deemed medically necessary and likewise recorded in the patient chart. All adverse events were classified according to the Common Terminology Criteria for Adverse Events (CTCAE), version 5.0 [21].

### 2.3. Clinical Assessment

Demographic characteristics included vital status, sex, age at the time of NF2 diagnosis, and age at the initiation of treatment. Clinical outcomes of interest comprised the treatment history associated with NF2 and vestibular schwannoma, diagnostic NF2 criteria [20], tumor burden and localization, family history of NF2, vestibulocochlear manifestations (hearing, tinnitus, vertigo, and/or imbalance), cranial nerve functionality, and peripheral neurology. Clinical evaluations were conducted at four distinct timepoints: upon initial presentation after referral, at the patient visit preceding start of bevacizumab treatment, at the conclusion of bevacizumab treatment, and at most recent patient visit during the extended follow-up period. In the event a patient underwent surgery or radiotherapy due to NF2-related disease progression, the last patient visit prior to re-intervention was deemed the termination point for follow-up after bevacizumab treatment.

The primary outcome of interest was hearing at the side of the target tumor at the end of bevacizumab treatment. Hearing was examined with pure tone audiometry (PTA) and the word recognition score (WRS). The PTA was measured as average hearing threshold at 0.5, 1, 2, and 3 kilo Hertz [22,23]. Hearing was defined as improved (≥10% increase in WRS), stable (no significant change in WRS), or worsened (≥10% decrease in WRS) in relation to hearing at the start of bevacizumab treatment. In case of 100% WRS at both times of measurement, a ≥10 decibel (dB) change in average PTA was considered significant. Secondary audiological outcomes of interest included subjective hearing and presence of tinnitus.

As NF2 patients often suffer from other symptoms than those related to the vestibular schwannomas, overall neurological symptoms were assessed to evaluate disease burden due to NF2-related lesions. We report subjectively improved, stable, or worsened overall symptoms related to NF2. Symptoms were divided in functioning of the trigeminal nerve, the facial nerve, and other cranial nerves (excluding the vestibulocochlear nerve) and peripheral neurology. Facial nerve function was graded according to the House Brackmann scale (HB) [24]. Other cranial nerves functions and peripheral neuropathy were collected using patient records.

### 2.4. Radiological Assessment

Radiological evaluations were limited to vestibular schwannomas. Target tumor characteristics were evaluated using routine clinical thin-slice (≤1.5 mm) gadolinium-enhanced T1-weighted Magnetic Resonance Images (MRIs) of the cerebellopontine angle [25,26]. The primary outcome of interest was radiologic response of the target vestibular schwannoma, defined as improved (≥20% decrease in extrameatal tumor volume), stable (no significant change in extrameatal tumor volume, >−20% to <20%), or worsened (≥20% increase in extrameatal tumor volume) at the end of bevacizumab treatment [25,27]. Secondary outcomes of interest were extrameatal tumor volume (cm^3^) and tumor growth rate. Tumor volume and growth rate were calculated using the largest linear dimensions in three planes (anterior–posterior; medial–lateral; cranial–caudal) from consecutive MRI scans [28,29,30]. Other intracranial lesions were not evaluated.

Similar to the clinical assessment, four standardized timepoints were used to assess radiologic characteristics of the target tumor: upon initial presentation after referral, at imaging preceding start of bevacizumab treatment, at the conclusion of bevacizumab treatment, and at most recent imaging during radiological surveillance after treatment.

### 2.5. Statistical Analysis

For statistical analyses, IBM SPSS Statistics for Windows (version 27.0. Armond, NY, USA: IBM Corp) was used. Median, range, frequency count, and percentage were used to report on non-normally distributed data. Standard descriptive statistics were used to describe patient and treatment characteristics. Missing, unused, and incorrect data were documented in the data collection and study report.

## 3. Results

### 3.1. Patient Characteristics

Seventeen patients were included in the present analysis. Demographics, treatment history, and disease characteristics are listed in Table 1. The median age was 39.1 (16.5–67.3) years at time of diagnosis and 48.1 (22.9–70.7) years at the start of initial bevacizumab treatment. Prior to bevacizumab therapy, two patients underwent partial surgical resection of the target vestibular schwannoma. Three patients received radiotherapy prior to bevacizumab but none of these on the tumor of interest. No patient was under treatment with bevacizumab at the time of data collection.

Four patients were excluded from the study for the following reasons: Three patients received bevacizumab at a different institute, and their data were incomplete. One patient passed away shortly after treatment was started, so follow-up data were not available. The cause of death in this case was unrelated to bevacizumab treatment.

### 3.2. Bevacizumab Treatment

Bevacizumab treatment was started a median of 57.5 (13.3–311.7) months after first presentation. Patients received a median dose of 7.5 (5.6–7.5) mg/kg every 3 (3.0–4.4) weeks. The duration of bevacizumab treatment ranged from 2.1 to 23.9 months, with a median of 7.1 months. The primary reason for bevacizumab therapy was progressive hearing loss in the majority of cases (59%). Other indications included symptoms related to other NF2-related schwannomas (35%) and target tumor size (6%).

Three patients deviated from the standard-of-care protocol. In two patients, bevacizumab dose was lowered to a maintenance dosage (orange bar, Figure 1). Patient 7 received a lower dosage of 5.0 mg/kg every 4 weeks for 2.9 months, while patient 12 received a dosage of 7.5 mg/kg every 6 weeks for 6.7 months. Patient 14 skipped one injection after 12 months to allow for a minor surgical intervention.

The most common reason to stop treatment was the achievement of the treatment goal(s): radiologic tumor stability or regression (29%), radiologic tumor stability or regression plus stable or improved hearing (24%), and improved symptoms (6%). Adverse events were the second most common reason to discontinue treatment (29%), and response failure was less frequently reported (6%). A single patient (6%) preferred to stop treatment despite the physician’s advice to continue.

For six patients, the end of treatment coincided with the last available patient visit. The eleven patients remaining were followed up after treatment for a median of 29.2 (2.7–52.4) months. After an extended time period, bevacizumab was reintroduced in four patients (Figure 1). Three patients also received bevacizumab a third time. Time between consecutive treatment courses varied from 7 to 64 months. Reasons to reintroduce bevacizumab were hearing loss (*n* = 2, 50%), vestibular complaints (*n* = 1, 25%), and tumor progression of other NF2-related schwannomas (*n* = 1, 25%).

### 3.3. Adverse Events

Out of seventeen patients, sixteen patients (94%) experienced at least one adverse event. One patient (6%) reported no treatment-related sequelae. A total of 50 adverse events were reported: 31 of grade 1 (62%), 17 of grade 2 (34%), and 2 of grade 3 (4%). Grade 4 toxicities or treatment-related deaths did not occur. The most commonly reported adverse events were hypertension (82%) and fatigue (29%). Five of fourteen patients with hypertension (36%) required antihypertensive medication during active treatment. An overview of all adverse events is shown in Table 2.

Treatment was discontinued (29%) due to an allergic dermatologic reaction after injection (*n* = 1), vertigo after injection (*n* = 1), edema of the limbs (*n* = 1), and a cluster of complications (*n* = 2). Patient 9 stopped bevacizumab after 24 months due to delayed medication-related adverse events. The majority of patients who had to stop bevacizumab (patients 4, 5, 11, and 13) experienced serious adverse events shortly after starting treatment.

All four individuals subjected to multiple courses of bevacizumab experienced more than one adverse event. During the second and third treatment courses, a total of 30 adverse events were documented: 21 classified as grade 1 (70%), 8 as grade 2 (27%), and 1 as grade 3 (3%). As observed in patients receiving only one course, the most common adverse event was hypertension. No instances of more severe toxicities were observed after multiple courses of bevacizumab.

### 3.4. Hearing Response

Between first presentation and the start of bevacizumab treatment, two patients experienced total hearing loss due to surgical resection of the target tumor. Consequently, hearing assessment in the target ear was feasible for fifteen of seventeen patients (Figure 1 and Figure 2). Three patients (patients 2, 9, and 12) already had maximum speech recognition at the start of treatment.

Improved hearing was detected in six of fifteen patients (40%), eight patients (53%) had stable hearing, and one patient (7%) experienced hearing loss. At the start of treatment, the median WRS was 74% (5–100) and PTA 57 (10–100) dB. After treatment, the median WRS improved to 82% (5–100), and the PTA slightly increased to 61 (10–88) dB. For patients 1, 4, 6, 15, and 17, the WRS increased ≥10%, but the PTA remained stable (>−10 dB to <10 dB). In patients 9 and 13, a stable WRS was reported, while the PTA improved from 41 to 21 dB and 100 to 73 dB, respectively. As patient 9 already had maximum speech recognition, hearing was regarded as improved based on the PTA. Both the WRS and PTA remained stable after treatment in patients 2, 3, 7, 8, 10, 12, and 16. Patient 14 experienced hearing loss, with a WRS of −10% and an increased PTA from 60 to 67 dB.

Audiometric follow-up after treatment discontinuation was available for eleven patients, with a median follow-up of 7.4 (2.7–63.3) months. Stable hearing was observed in nine of eleven patients (82%), while two patients (18%) experienced hearing loss. These two patients and two others underwent a second course of bevacizumab. Following the second treatment course, one patient experienced hearing improvement and three patients had stable hearing. Hearing loss during surveillance after the second treatment course prompted three patients to resume bevacizumab treatment for a third time. In two of three patients hearing stabilized, while in the third patient, hearing loss progressed.

### 3.5. Radiologic Response

One patient was excluded as only repeated Computed Tomography (CT) images were available. Radiologic tumor response could therefore be evaluated in sixteen patients (Table 3) (Figure 1 and Figure 3). Two patients had a partial resection of the target vestibular schwannoma before treatment with bevacizumab. Their tumor characteristics prior to bevacizumab were disregarded, but the radiologic responses of the tumor residue following bevacizumab treatment were included.

Five patients (31%) showed extrameatal tumor regression, while eleven (69%) showed a stable tumor. No patient experienced tumor progression (≥20%) on active treatment. Median extrameatal tumor volume shrunk from 1.24 (0.06–12.67) cubic centimeters (cm^3^) at the start of treatment to 1.15 (0.0–10.52) cm^3^ after treatment. At the start of treatment, extrameatal tumor growth was 43% (−61~+816) overall or 15% (−5~+150) annually (median follow-up 55.9 months; 16.2–218.5). During treatment, tumor growth lowered to −12% (−100~+6) overall or −13% (−161~+5) annually (median follow-up 12.6 months; 4.3–28.8).

Data on radiologic surveillance after bevacizumab were available in twelve patients (median follow-up 19.3 months; 4.0–52.4). Continued tumor regression was detected in one patient (9%), four patients (36%) had stable target tumors, and six patients (55%) showed tumor progression.

After the second course of bevacizumab, one patient experienced tumor regression, while three patients had stable tumors. One vestibular schwannoma remained stable for at least 10.6 months. This patient did not receive another course of bevacizumab. The other three patients received a third reintroduction of bevacizumab. In between the second and third treatment course, all patients experienced tumor progression. During the third course of bevacizumab, tumor regression, stabilization, and growth were observed in one patient each.

### 3.6. Symptomatic Response

Symptoms could be evaluated in all seventeen patients (Figure 1). At the start of treatment, all patients (*n* = 17, 100%) reported vestibulocochlear complaints: sixteen patients (94%) experienced hearing loss, fourteen patients (82%) reported tinnitus, and ten patients (59%) suffered from imbalance. Paresis of the facial nerve was present in six patients (35%). Of these, three patients experienced a mild paresis (HB3; 50%), one patient a severe paresis (HB5; 17%), and two a total paralysis (HB6; 33%). Neuropathy of the trigeminal nerve and other cranial nerves was present in four (24%) and six (35%) patients, respectively. Seven patients (41%) experienced peripheral neuropathy.

Following treatment with bevacizumab, seven patients (41%) experienced an improvement of symptoms, eight patients (47%) maintained stable symptoms, and two patients (12%) experienced a deterioration in symptoms. Improved symptoms included hearing (*n* = 5), tinnitus (*n* = 1), balance (*n* = 2), vision (*n* = 2), and/or peripheral nerve functioning (*n* = 2). One patient reported a new tingling sensation of the hand but also reported improved hearing and vision and was therefore categorized as an overall improvement of symptoms. Similarly, another patient experienced increased unsteadiness but likewise demonstrated improved hearing and vision and was consequently considered to have experienced an overall improvement. Both patients who reported symptomatic deterioration experienced greater imbalance.

Clinical surveillance after discontinuation of bevacizumab was available in eleven patients, with a median time of 29.2 (2.7–52.4) months. After treatment was stopped, no patient experienced further improvement of symptoms, three patients (27%) experienced stable symptoms, and eight (73%) had increased symptomatology. All four patients that received a second or third treatment course had experienced worsened symptoms after initial treatment discontinuation. Following the second treatment course, one patient experienced symptom improvement, one experienced stable symptoms, and two patients experienced worsened symptoms. Three patients, one with stable symptoms and two with worsened symptoms, received a third course of bevacizumab. Thereafter, symptoms improved in one patient, remained stable in one, and worsened in another.

## 4. Discussion

An overall positive response to bevacizumab was found in the present cohort of NF2 patients, using a treatment regimen of 7.5 mg/kg bevacizumab every three weeks. Notably, hearing preservation was observed in 93% of the patients (improved in 40%; stable in 53%). Similar responses were found for tumor size, as 31% of the target tumors exhibited regression, and 69% remained stable. In addition, NF2-related symptoms improved in 88% of the patients.

Comparable findings were reported in a recent systematic review including 200 NF2 patients treated with bevacizumab [18]. The pooled data in the study indicated hearing improvement in 45% and stabilization in 55% of the patients. Moreover, tumor regression was observed in 38%, while stabilization occurred in 53% of the patients. It is important to note, however, that the present cohort differed from the pooled population in the systematic review. In our study, a higher median age and a shorter treatment duration were found. Additionally, the systematic review included two studies focused on pediatric patients, which was beyond the scope of the present study. On the other hand, a greater variability in dosage regimens was reported in the systematic review. A multicenter, prospective study of 22 NF2 patients (15 adult; 7 pediatric) also evaluated the efficacy and toxicity of bevacizumab for vestibular schwannoma at a higher dosage (10 mg/kg every two weeks for six months) [31]. In the study, hearing improved in 41% of the patients. Tumor regression was observed in 32%.

In our study cohort, adverse events, particularly hypertension (82%) and fatigue (29%), were frequently observed. The incidence of hypertension was higher than in previous studies (33–58%) [18,19,31,32]. A possible explanation may be the higher median age in our cohort and differences in dosage regimens between studies, as both factors have been found to be predictors of developing hypertension during bevacizumab treatment [19]. The frequency of measuring blood pressure during treatment may also account for differences between studies. Even so, grade 3 hypertension was infrequent (12% of patients) and aligned with the rates reported in other studies [11,15,19,33].

Fatigue stands out as a frequently documented complication arising from the administration of bevacizumab. The incidence in our study (29%) was comparable to the previously reported incidence (23–64%) [11,15,19,31,33].

The prevalence of proteinuria was relatively low in the present cohort. It was observed in only 6% of the patients during the initial treatment with bevacizumab. Other studies have reported higher rates (43–62%) [18,19]. It has been hypothesized that proteinuria is a late complication of bevacizumab treatment, with intervals between start of treatment and proteinuria onset of 10.7 to 23.7 months [19,32]. The lower rate of proteinuria in our study may therefore be explained by the shorter treatment duration of 7.1 months. An alternative explanation could be differences in execution of urinalysis. In our institute, the evaluation of proteinuria during treatment with bevacizumab was performed at variable time intervals.

Early termination of bevacizumab treatment occurred in 29% of patients in the present study, which falls within the upper range of dropout rates observed in other studies (9–28%) [11,13,15,33]. The notable rate of treatment discontinuation underscores that bevacizumab is not well-tolerated by a significant number of patients. The nature of toxicity leading to treatment discontinuation varies across studies and appears to be highly patient-specific. Notably, the way in which treatment toxicities are reported is diverse and often lacks standardization, with the Common Terminology Criteria for Adverse Events (CTCAE) being a frequently employed classification system [21]. However, the use of other national or international guidelines further complicates the comparison of adverse event rates between studies. Careful monitoring of adverse effects, according to a standardized format, should remain a relevant treatment parameter in future studies on bevacizumab for NF2 patients.

In the present study, only two patients were partly treated with bevacizumab under a maintenance regimen. Both patients showed a stable response despite the variation from the standard treatment protocol. In a small case series, patients restarted bevacizumab at a reduced dosage (2.5 mg/kg every two weeks), resulting in stable hearing and tumor size during the maintenance phase, though limited data were available [34]. Other studies have also used a maintenance regimen to enable long-term treatment of NF2 patients that showed a stable clinical response, but an optimal dosage threshold remains to be determined [15,31,35].

Limited data exist on NF2 patients resuming bevacizumab after an initial course, and the available literature mainly addresses treatment breaks due to factors such as toxicity, surgical interventions, or patient preference, with most breaks lasting only a few months. Insight into the within-subject variability of bevacizumab efficacy and toxicity after extended treatment gaps is scant. Extended breaks (≥3 months) have been associated with tumor (re)growth and hearing decline, consistent with our findings [15,33,34]. We found notable variability in the response after reintroduction of bevacizumab, encompassing both between-subject and within-subject differences in efficacy and toxicity during consecutive treatment courses (Figure 1). As with other associated symptoms, hearing may restabilize or improve after the restart of bevacizumab therapy, but further progression is also observed. The same holds true for tumor progression. This implies that successful restart of bevacizumab is feasible if symptoms or tumor progression recur after prolonged treatment stops; however, previous positive effects do not reliably predict future treatment success.

The present study has several strengths. Firstly, the data analysis provides an extensive longitudinal examination of NF2 patients receiving bevacizumab. This study extends beyond assessing the efficacy and toxicity of bevacizumab during active initial treatment, incorporating a clinical analysis of NF2 patients from their initial presentation at the expert center through the first administration of bevacizumab, spanning over 4.8 years. Post treatment, patients continued to be monitored for a median duration of 2.5 years. Secondly, in addition to audiometric and radiologic follow-up, symptoms were included in the analysis. Thirdly, this study also assessed the effect of sequential courses of bevacizumab in patients that resumed treatment after extended treatment gaps.

Study limitations include the retrospective design and the small sample size, inherent to the rarity of this disease. This is illustrated by a recent systematic review on this topic, in which only 10 patient cohorts totaling 200 patients could be identified, with a median number of 16.5 included patients per cohort.

Tumor volume measurements were calculated using three linear dimensions. These are therefore an estimate of the actual tumor volumes. Additionally, variations in the timing of follow-up measurements for hearing, radiology, and symptoms among patients introduce variability. This makes it difficult to draw conclusions about the timing and durability of the response after treatment. In the present study, four general timepoints were used to assess outcome measurements, yet the clinic’s follow-up regimen has evolved over time. Nowadays, our patients are followed up every six months.

The management of NF2 remains challenging, despite successful therapeutic interventions. Bevacizumab provides an effective alternative to surgery or radiotherapy for vestibular schwannoma and related symptoms. However, its effect on target tumors is variable. Moreover, different bevacizumab regimens and surveillance protocols are used across institutes, and the effect of dosage on reported efficacy and toxicity remains uncertain. It appears that lower dosages of bevacizumab result in a similar treatment efficacy compared to higher dosages, while toxicity may be less substantial [18]. This study identified an overall positive response to 7.5 mg/kg bevacizumab every three weeks, consistent with these findings [18]. Nevertheless, adverse effects were common, and a quarter of all NF2 patients eligible for bevacizumab discontinued treatment prematurely due to toxicity. Therefore, we carefully suggest an updated treatment dosage of 7.5 mg/kg every three weeks for three months, followed by a maintenance dosage of 5 mg/kg every four weeks for three months.

Limited attention is still given to the overall disease burden and patient-reported outcomes. While hearing and tumor volumetrics are paramount, symptoms related to NF2-related tumors and quality of life in relation to different management options are often overlooked. Given the chronic and progressive nature of NF2, patients often undergo multiple and repeated interventions over time. While some respond favorably to multiple courses of bevacizumab, the response is not universal. Both tumor and clinical responses show noticeable between-subject and within-subject variability. Predicting the response to bevacizumab treatment would be valuable for developing a more effective and personalized treatment plan. Future studies in a multicenter, randomized format with larger cohorts would be ideal to evaluate the effectiveness of different treatment regimens and gain insight into factors that predict the outcome. Still, these may remain elusive due to the rarity of the disease and the inherent complexities of study design and execution. Reporting experiences with large case series such as these will therefore continue to be important in furthering our understanding of the disease and the treatment.

## 5. Conclusions

The majority of *NF2*-related schwannomatosis patients treated with bevacizumab had an overall positive therapeutic response. After treatment, all target vestibular schwannomas showed tumor stabilization or regression, and the large majority of patients experienced improvement or stabilization of hearing and other NF2-related symptoms. However, adverse events were common, and a quarter of all patients treated with bevacizumab discontinued treatment prematurely due to toxicity. Therefore, careful consideration of treatment benefits and risks is warranted to provide patients with an appropriate management strategy.

## Figures and Tables

**Figure 1 cancers-16-01479-f001:**
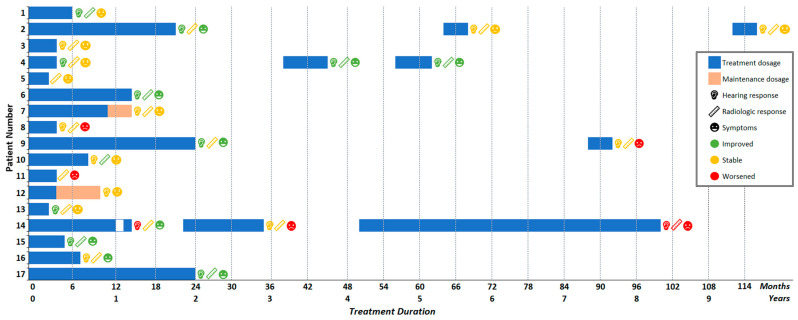
Treatment regimens and response of individual patients in the cohort. Duration and dosage of treatment is indicated by bar length and coloring, respectively. At the end of each treatment course, hearing, radiologic, and symptomatic response were recorded. Symbols indicate the outcome of interest, coloring indicates an improved, stable, or worsened response. Four patients received more than one bevacizumab treatment course. Time interval between consecutive treatments corresponds to the white area between the horizontal bars.

**Figure 2 cancers-16-01479-f002:**
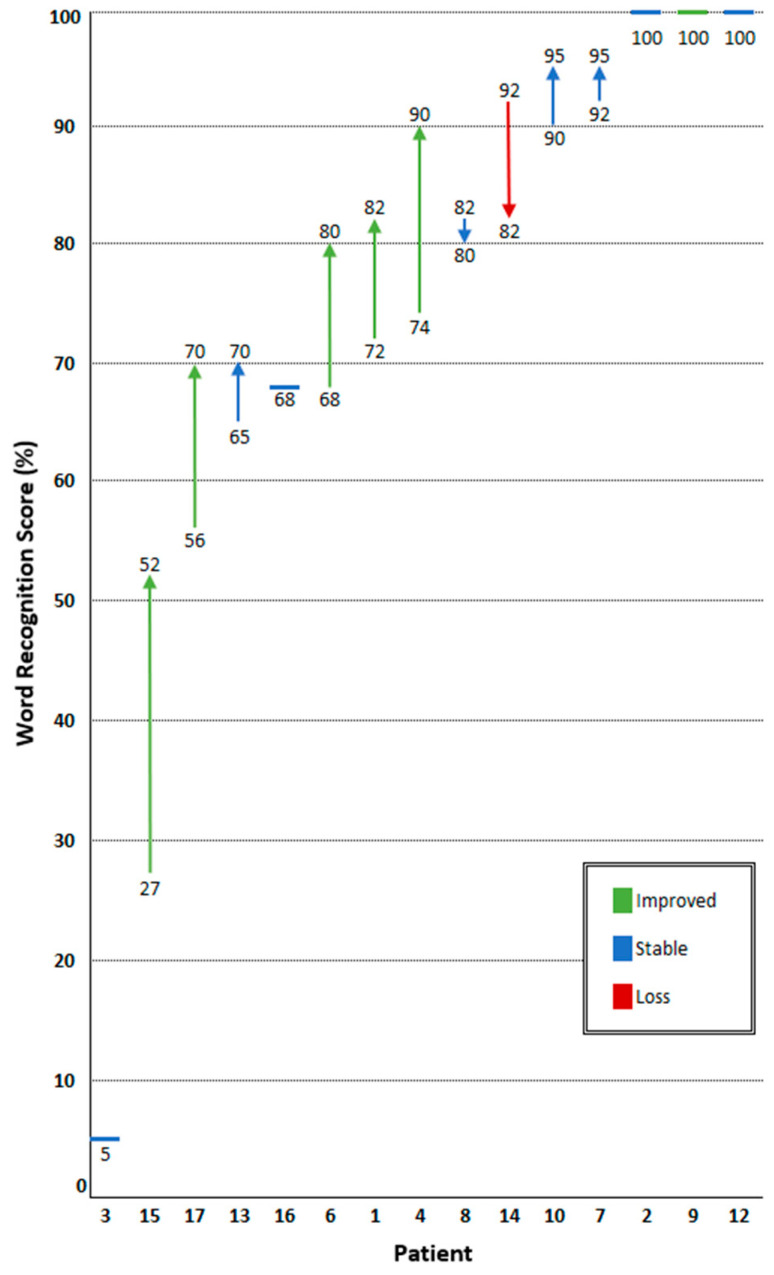
Change in maximum word recognition score (WRS) after treatment with bevacizumab. Patients are represented on the horizontal axis. Individual WRS before and after bevacizumab are stated above and below the arrows. Percentage change in WRS is represented by the length of the arrow, with the direction representing hearing improvement or loss. Six patients (40%) experienced improved hearing, eight patients (53%) had stable hearing, and one patient (7%) experienced hearing loss. Three patients (patients 2, 9, and 12) already had maximum speech recognition at the start of treatment.

**Figure 3 cancers-16-01479-f003:**
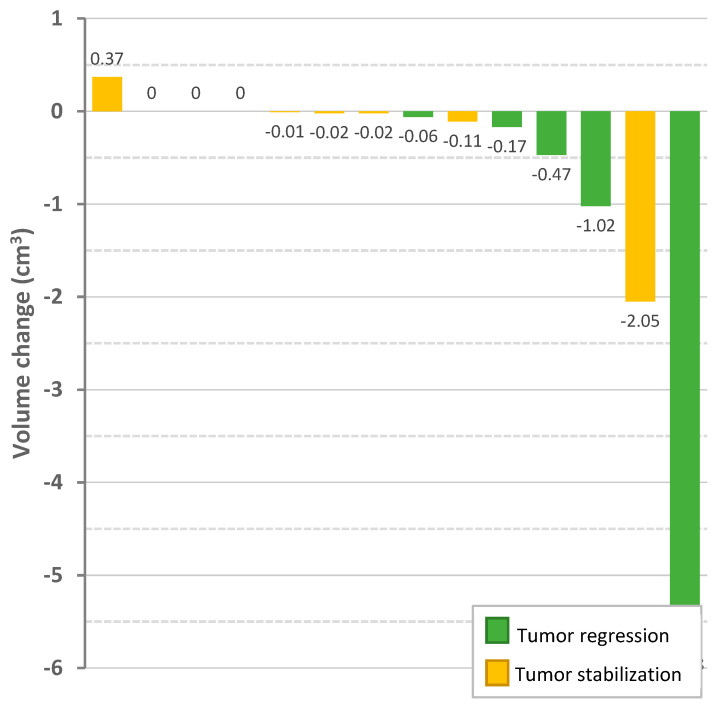
Change in absolute extracanicular volume (cm^3^) of vestibular schwannomas after bevacizumab treatment. Vertical bars represent the target vestibular schwannoma of a patient. Coloring of the bars indicate the corresponding tumor response: green for tumor regression (volume change ≥20%), yellow for stable tumor (volume change between −20% and 20%). Three target vestibular schwannomas are not shown: one tumor was confined to the internal auditory canal, two other tumors were not followed up with MRI at the end of treatment.

**Table 1 cancers-16-01479-t001:** Demographics and disease characteristics of *NF2*-related schwannomatosis patient cohort.

Patient Characteristics		Patients, *n* = 17
Alive, *n* (%)		16 (94.1)
Age, years, median (range)	Time of diagnosis Start of treatment	39.1 (16.5–67.3) 48.1 (22.9–70.7)
Female, *n* (%)		9 female (52.9)
Family history of NF2, *n* (%)		2 (11.8)
Previous treatment, yes, *n* (%)	Total	11 (64.7)
	Surgical resection Target vestibular schwannoma Non-target tumor(s)	10 (58.8) 2 (11.8) 10 (58.8)
	Radiotherapy Target vestibular schwannoma Non-target tumor(s)	3 (17.6) 0 (0.0) 3 (17.6)
NF2 classification, *n* (%) ^1^	Class 1 Class 2 Class 3	15 (88.2) 0 (0.0) 2 (11.8)
Tumor localization, *n* (%)	Unilateral vestibular schwannoma Bilateral vestibular schwannoma	2 (11.8) 15 (88.2)
	Non-vestibular schwannoma Central nervous system Peripheral nervous system	12 (70.6) 12 (70.6) 3 (17.6)
Total tumor count, median (range)		12 (2–40)

^1^ According to the updated diagnostic criteria [20]. Legend: *n* = number of patients; NF2 = *NF2*-related schwannomatosis.

**Table 2 cancers-16-01479-t002:** Adverse events and laboratory abnormalities during treatment with bevacizumab, graded according to Common Terminology Criteria for Adverse Events (version 5.0) [21].

Adverse Event ^1^	Patients, *n* (%)	Grade 1, *n* (%)	Grade 2, *n* (%)	Grade 3, *n* (%)
Total	16 (94.1)	13 (76.5)	12 (70.6)	2 (11.8)
Hypertension	14 (82.4)	3 (17.6)	9 (52.9)	2 (11.8)
Fatigue	5 (29.4)	5 (29.4)		
Elevated liver enzymes	3 (17.6)	3 (17.6)		
Diarrhea	2 (11.8)	2 (11.8)		
Dry skin	2 (11.8)	2 (11.8)		
Epistaxis	2 (11.8)	2 (11.8)		
Headache	2 (11.8)	1 (5.9)	1 (5.9)	
Hyperkalemia	2 (11.8)	1 (5.9)	1 (5.9)	
Impaired wound healing	2 (11.8)	2 (11.8)		
Infection	2 (11.8)	2 (11.8)		
Irregular menses ^2^	2 (22.2)	1 (11.1)	1 (11.1)	
Myalgia	2 (11.8)	2 (11.8)		
Nausea	2 (11.8)	1 (5.9)	1 (5.9)	
Allergic dermatologic reaction	1 (5.9)	1 (5.9)		
Dysphonia	1 (5.9)	1 (5.9)		
Edema of limbs	1 (5.9)		1 (5.9)	
Erectile dysfunction	1 (5.9)	1 (5.9)		
Hyperglycemia	1 (5.9)	1 (5.9)		
Proteinuria	1 (5.9)		1 (5.9)	
Thrombocytopenia	1 (5.9)	1 (5.9)		
Vertigo	1 (5.9)		1 (5.9)	

^1^ Number of patients and percentages taken from entire cohort. ^2^ Based on total of female patients (*n* = 9). Legend: *n* = number.

**Table 3 cancers-16-01479-t003:** Radiological characteristics of the target vestibular schwannomas before, during, and after treatment with bevacizumab.

Radiologic Characteristics		*n* =
First presentation		16
Extrameatal extension, *n* (%)	13 (81.3)	
Extrameatal volume, cm^3^, median (range)	2.07 (0.09–24.45)	
Start of bevacizumab treatment		16
Extrameatal extension, *n* (%)	14 (87.5)	
Extrameatal volume, cm^3^, median (range)	1.24 (0.05–12.67)	
Relative change, % (range)	43 (−61–816)	
End of bevacizumab treatment		16
Extrameatal extension, *n* (%)	13 (81.3)	
Extrameatal volume, cm^3^, median (range)	1.15 (0.0 ^1^–10.52)	
Relative change, % (range)	−12 (−100 ^1^–6)	
Most recent radiologic surveillance		12
Extrameatal extension, *n* (%)	11 (91.7)	
Extrameatal volume, cm^3^, median (range)	1.52 (0.03–8.25)	
Relative change, % (range)	27 (−22–275)	

^1^ Full regression of extracanicular tumor component. Legend: *n* = number of patients, max. = maximum, cm = centimeters.

## Data Availability

The data presented in this study are available on request from the corresponding author. The data are not publicly available due to privacy restrictions.

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
