# Peer review of "Bevacizumab Treatment for Patients with NF2-Related Schwannomatosis: A Single Center Experience"

_cancers, 2024, doi:10.3390/cancers16081479_

Round 1

Reviewer 1 Report

Comments and Suggestions for Authors

Dear Editors, Dear Authors,

Thank you for the opportunity to review this work.

The authors performed a retrospective analysis investigating the outcome of hearing and growth control with Off-Label Treatment with Bevacizumab in 17 NF2 patients.

All patients included were ≥ 18 years of age at the time of diagnosis and received a dose of 7,5 mg/kg bw every 3 weeks. The median age was 39.1 (16.5 – 67.3) years at the time of diagnosis and 48.1 (22.9 – 70.7) years at the start of initial bevacizumab treatment. The duration of bevacizumab treatment ranged from 2.1 to 23.9 months, with a median duration of 7.1 months.

The hearing was predominantly evaluated by WRS (rather than PTA) and volume changes were determined by diameter (linear) based measurements. The authors detected a 40%/53% rate in hearing improvement/stabilization and a radiological regression/stabilization occurred in 31%/ 69%.

There are numerous studies describing the efficacy of off-label therapy with Bevacizumab in NF2-associated vestibular schwannomas. Some important findings have already been obtained, such as a dose recommendation of 2.5 mg/kg bw every 2 weeks as an initial dose (has similar efficacy to 5 mg in adults, see Farschtschi et al. doi: 10.1007/s00405-015-3604-y ) and, in particular, the finding that children and adolescents respond significantly worse to bevacizumab than adults (Morris et al. 10.1093/nop/npv065, Gugel et al. 10.3390/cancers11121862).  Furthermore, smaller tumors respond less well than large tumors (Guge et al. 10.3390/cancers11121862).

In addition, many studies also describe the long-term course of treatment and the side effect profile well.

The current study is well described but does not provide any new findings.

In particular, the study did not address the problem of the difference in response (children/adolescents vs. adults) in more detail (the current cohort is comparatively “very old” for an NF2 cohort). A positive response from the current cohort was therefore expected. An observation period under BVC of 2-23 months is also short.

Crucial is whether the observed response of the current cohort can be sustained then. This usually becomes clear only after 1-2 years. The effect after 6 months does not reflect the long-term response, and we know that tumors also respond differently under long-term treatment (Gugel et al. 10.3390/cancers11121862). Therapy often needs to be administered for longer than 24 months, so the current study describes a short period.

Regarding the limitations. It is certainly correct and necessary to point out limitations. It is clear that there are smaller cohorts in NF2; however, these cohorts would significantly benefit from long-term data (> 2 years). Especially if important new insights emerge as a result. The aspect of the non-performed software-based tumor volumetry, which is, of course, more time-consuming but our gold standard for monitoring these tumors, is a significant limitation. Linear-based volume calculations are simply too inaccurate, and thus, the radiological response described in the study cannot be utilized. Consequently, this study cannot be compared with 3D-based methods, which are significantly more accurate.

It is also not clear how the volume was then calculated (which sequences/layers were used, which formula?). How many MRI data sets were available per patient or/and in total?

I also cannot properly comprehend why primarily the WRS and only secondarily the PTA were used to assess hearing. How much audiological data were available? How often were MRIs and audiometry performed during the short period of BVC therapy?

Author Response

Please find our reply to Reviewer 1 in the Word-document attached.

Reviewer 2 Report

Comments and Suggestions for Authors

Douwes et al. presented their findings regarding clinical outcomes and adverse reactions following bevacizumab treatment in NF2 patients. Bevacizumab, a monoclonal antibody, inhibits angiogenesis in tumors and their surrounding areas via the VEGF pathway. Several additional reports on the use of bevacizumab to treat patients with neurofibromatosis type 2 have also been published. The study examined changes in hearing function, neurological deficits, and adverse reactions in patients treated with bevacizumab, demonstrating positive results with improved hearing and tumor size reduction.

Several key concerns have been identified:

1.     The abstract structure requires revision, favoring a more literature-focused approach over a columnar format for publication.

2.     A primary issue with the study is the lack of novel findings and the relatively small sample size (n = 17).

3.     Evoked potential examinations are crucial for evaluating auditory brainstem and peripheral-to-central nerve conduction. In retrospectively collected data, including auditory brainstem, somatosensory, and motor evoked potentials is essential for assessing functional deficits during treatment.

4.     In addition to bilateral acoustic Schwannomas, NF2 patients commonly exhibit intracranial meningiomas and spinal cord gliomas. If whole-body MR studies are challenging, evaluating MR imaging studies of the central nervous system, including the spinal cord, is essential for assessing tumor burden and progression.

5.     Recording "symptomatic response" post-treatment appears to rely on patient-reported outcomes. Objective evidence such as changes in nerve conduction parameters or evoked potential studies would provide more compelling information.

A few minor concerns are suggested as follows:

1)      Line 42: ……..nervous system à ……nervous systems

2)      Line 44: Please leave out the word, “invalidating”.

3)      Line 46: Please leave out the word, “more”.

4)      Line 54~55: “….6% increased risk of benign nervous system tumors becoming malignant or new primary tumors developing in irradiated tissues” is suggested to reword into “ … 6% increase risk of malignant transformation or newly tumor growth on the irradiated tissues”.

5)      Line 56~57. The sentence needs to be rephrased.

6)      Line 68~70. The authors need to point out the new findings as a synopsis for the introduction of this study. 

7)      Line 166~169. The four patients were included in the total 17 patients, or not? If not, I would suggest to leave them out.

8)      The presentation of figure 1 looks too complicated for the readers.

9)      Line 201. “Eleven remaining patients were……….” can be changed into “The rest eleven patients were…………..”.

10)   It needs a color legend for Figure 3.

11)   Line 276. The “cm3” needs to be changed into “cm3”.

12)   Line 277. To make it clear, the “(-61 - +816)” needs to be changed into “(-61 ~ +816)”. The between sign “-“may be confused with the negative sign.

13)   Line 348. “……with rates reported in other studies” can be changed into “…….with the rates reported in other studies”.

14)   Line 361~362. The sentence needs to be rephrased.

15)   Line 411~412. The claim of this sentence needs a reference.

Comments on the Quality of English Language

The quality of English is well-above the average. 

Author Response

Please find our reply to Reviewer 2 in the Word-document attached. 

Reviewer 3 Report

Comments and Suggestions for Authors

The manuscript represents an interesting clinical analysis of type 2 schwannomatosis and the use of bevacizumab. The description of the patients and treatment is well defined as the rationale. I have some general  questions concerning the results and several minor suggestions.

One question refers to the monoclonal doses. It states that the dose is every three weeks for at least six months; however, toxicity by the treatment was observed earlier in 5 patients. It is understandable that the first line of treatment for the disease should the monoclonal, but close to 30 % did not reach the minimum goal. What other therapeutic options do the authors suggest may be useful in the future?

Other issues. It is unclear if patients were treated before the use of monoclonal. Figure 1 is confusing please simplify the legend.

Patients 4 and 15 were special and very rare. What was the rationale for the treatment scheme of patient 4? Is the age or gender a factor in the response to the treatment? Was hypertension due to the treatment solved after discontinuing the treatment? 

Figure 3 is the most interesting, but it requires a summary: 6 individuals with improvement (one of them was already at 100), 8 had no improvement (2 were already at 100), and 1 had hearing loss. These numbers should be in the conclusions. Moreover, the limitations of the secondary effects upon treatment should be considered a relevant point for further research.

Comments on the Quality of English Language

Minor grammatical mistakes were encountered

Author Response

Please find our reply to Reviewer 3 in the Word-document attached.

Round 2

Reviewer 1 Report

Comments and Suggestions for Authors

Thank you very much for answering my points. Unfortunately, this has not substantially improved the manuscript, as the data itself is too thin compared to the previous literature and no new relevant findings have emerged as a result.
The authors described that the strength lies in the description of multiple phases or pauses in bevacizumab therapy. However, this does not emerge from the data analysis, in which the focus should also be directed towards this. However, these findings/evaluations would not be new either; there are already excellent detailed long-term data in this area that describe growth and hearing under several periods/gaps of bevacizumab.

Author Response

Please find our reply to Reviewer 1 in the document attached.

Reviewer 2 Report

Comments and Suggestions for Authors

I have no more comments.

Author Response

We thank Reviewer 2 for their valuable time and effort.